# GeoBS: Information-Theoretic Quantification of Geographic Bias in AI Models

## Abstract

The widespread adoption of AI models, especially foundation models (FMs), has made a profound impact on numerous domains. However, it also raises significant ethical concerns, including bias issues. Although numerous efforts have been made to quantify and mitigate social bias in AI models, **geographic bias** (in short, geo-bias) receives much less attention, which presents unique challenges. While previous work has explored ways to quantify geo-bias, these measures are *model-specific* (e.g., mean absolute deviation of LLM ratings) or *spatially implicit* (e.g., average fairness scores of all spatial partitions). We lack a **model-agnostic, universally applicable, and spatially explicit** geo-bias evaluation framework that allows researchers to fairly compare the geo-bias of different AI models and to understand what spatial factors contribute to the geo-bias. In this paper, we establish an **information-theoretic framework for geo-bias evaluation**, called **GeoBS** (**Geo**-**B**ias **S**cores). We demonstrate the generalizability of the proposed framework by showing how to interpret and analyze existing geo-bias measures under this framework. Then, we propose three novel geo-bias scores that explicitly take intricate spatial factors (multi-scalability, distance decay, and anisotropy) into consideration. Finally, we conduct extensive experiments on 3 tasks, 8 datasets, and 8 models to demonstrate that both task-specific GeoAI models and general-purpose foundation models may suffer from various types of geo-bias. This framework will not only advance the technical understanding of geographic bias but will also establish a foundation for integrating spatial fairness into the design, deployment, and evaluation of AI systems.

## 1 Introduction

Recent years have witnessed a major paradigm shift in the Artificial Intelligence (AI) domain from task-specific models to foundation models (FMs) (Bommasani et al., 2021). However, the widespread adoption of FMs also raises significant ethical concerns. A major challenge is bias (Gordon & Desjardins, 1995; Gianfrancesco et al., 2018), which refers to systematic disparities or tendencies in AI models that lead to unfair or prejudiced outcomes in terms of gender, race, religion, nationality, etc. AI model bias is a well-documented issue that can lead to serious ethical consequences, with notable examples such as Google Photo App's racist blunder of misclassifying African American people as gorillas (News, 2015). Due to the large training datasets and model sizes, FMs are more prone to inheriting and amplifying these biases (Zhang et al., 2022; Touvron et al., 2023), thus having the potential to cause a huge negative impact on society (Kamiran & Calders, 2012; Dhamala et al., 2021; Hardt et al., 2016; Parrish et al., 2021; Gallegos et al., 2024). Extensive efforts have been made for social biases evaluation and mitigation in terms of gender, religion, race, color, sexual orientation, etc, in AI models, including FMs, by developing bias evaluation frameworks, benchmark datasets (Nangia et al., 2020; Nadeem et al., 2021; Sheng et al., 2019; Dhamala et al., 2021; Gallegos et al., 2024), and debiasing methods (Yang et al., 2023; Xie et al., 2022). However, few efforts have been devoted to quantifying and addressing **geographic bias** of AI and FMs, which demonstrate unique characteristics and require novel bias quantification methods and debiasing approaches (Liu et al., 2022; Faisal & Anastasopoulos, 2023; Manvi et al., 2024).

Most previous work interprets it as a subclass of model bias – *a phenomenon that an AI model performs systematically differently across geographic regions beyond reasonable random fluctuations*

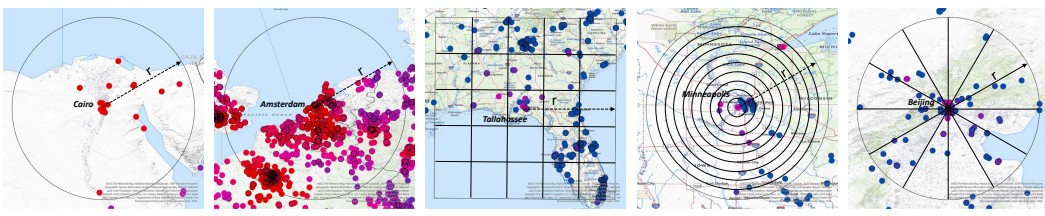

(a) Unmarked SSI    (b) Marked SSI    (c) Scale-Grid SRE   (d) Distance-Lag SRE   (e) Direct.-Sec. SRE

Figure 1: Illustrations of the 5 types of geo-bias where "Direct.-Sec. SRE" stands for Direction-Sector SRE. The dataset is fMoW (Christie et al., 2018). The evaluated models are Sphere2Vec-sphereC (Mai et al., 2023) for Figure (a), (b), and NeRF (Mildenhall et al., 2020) for Figure (c), (d), (e). Dots represent an evaluated data point. Darker red dots indicate worse model performances at this location.

(Liu et al., 2022; Xie et al., 2022; Wu et al., 2024). Despite the recent advancement in geo-bias research, we identify two major research gaps:

1. **Existing geo-bias metrics are often developed ad-hoc for specific models/tasks, and lack a systematic framework.** For example, Manvi et al. (2024) proposed a Spearman's $\rho$ bias score, which is only applicable to certain LLM zero-shot prompting tasks.

2. **Existing geo-bias metrics are implicit and lack a direct connection to their spatial implications**, i.e., what spatial factors (distance, direction, scale, etc.) stand behind the observed bias. For example, Xie et al. (2022) proposed to randomly partition the space and use the model's average performance difference over all possible partitioning as the geo-bias metric, which is hard to interpret because it mixes different types of geo-bias.

In this paper, we propose to bridge the aforementioned gaps with the help of classic spatial statistics and information theory by establishing an **information-theoretic framework for geo-bias evaluation**, called **GeoBS**. From a perspective of spatial point pattern analysis, a set of geolocations associated with corresponding model performance metrics forms a distribution on the Earth's surface, which can be treated as *spatial point patterns* (SPP) (Illian et al., 2008; Boots & Getis, 2020). The properties (e.g., Gaussian v.s. Poisson) and strengths (e.g., 10% Gaussian v.s. 90% Gaussian) of such patterns can help us categorize existing geo-bias metrics and design novel ones. For instance, if we define a Gaussian distribution as the "homogeneous" reference pattern that exhibits no geo-bias, then a statistical distance such as KL-divergence or Wasserstein distance effectively measures *how far the observed SPP deviates from the predefined homogeneity*, which may serve as a valid geo-bias metric.

To concretely demonstrate the power and generalizability of our framework, we first prove that two recently proposed and widely adopted geo-bias scores, Unmarked SSI and Marked SSI (Wu et al., 2024), can be clearly interpreted and organically integrated into our geo-bias framework. Then we propose three novel geo-bias scores under the guidance of this framework, which are able to differentiate and quantify geo-bias related to different spatial factors such as multi-scalability, distance decay, and anisotropy. Specifically, the Scale-Grid Spatial Relative-Entropy (SRE) score (Figure 1c) considers the multi-scale heterogeneity, i.e., at what spatial scales the low-performance points concentrate. The Distance-Lag SRE (Figure 1d) considers the distance-decay effect, i.e., whether the model performance changes as distance increases. The Direction-Sector SRE (Figure 1e) considers directional heterogeneity/anisotropy, i.e., whether the model performs differently in different directions. These scores allow us to locate the intricate spatial factors behind the observed geo-bias and better target potential solutions. All five geo-bias scores are conceptualized in Figure 1. In summary, the major contributions of this paper are:

1. We propose a theoretical framework to evaluate geo-bias from the perspective of information theory, which allows us to systematically categorize and interpret geo-bias.

2. We draw connections between spatial point pattern analysis with information theory, which allows us to design model-agnostic and spatially explicit geo-bias scores.

3. We demonstrate that our theoretical framework can successfully interpret existing geo-bias scores (e.g., Unmarked SSI and Marked SSI) and propose three novel geo-bias scores (Scale-Grid SRE, Distance-Lag SRE, Direction-Sector SRE) that can explicitly capture the intricate spatial factors behind observed geo-bias.

4. We extensively evaluate the geo-bias of both task-specific GeoAI models and task-agnostic foundation models. It is shown that both model groups demonstrate substantial geo-bias, and it is

important to use the spatially explicit geo-bias scores to interpret their underlying spatial factors behind geo-bias because many models show geo-bias of mixed types.

5. We implement a plug-and-play Python package called **GeoBS** for efficiently computing the five geo-bias scores. It will facilitate researchers to promptly check and report the geo-bias of their models, promoting spatial fairness in the community.

## 2 RELATED WORK

AI models, including task-specific GeoAI and general-purpose foundation models, often perform differently across various spatial contexts (Xie et al., 2022; Manvi et al., 2024; Faisal & Anastasopoulos, 2023; Wu et al., 2024). Such bias may lead to or even exacerbate inequities in resource allocation, social disparities, and vulnerabilities in resilience and sustainability (Xie et al., 2022) raising ethical concerns (Nelson et al., 2022). The objective of geo-bias metrics is to quantify geospatial bias, which are inherently linked to geographic locations or spatial distributions of data samples (Hay, 1995; Xie et al., 2022).

There has been extensive research on improving fairness in AI using pre-processing (Jo & Gebru, 2020; Steed & Caliskan, 2021), in-processing (Kamishima et al., 2011; Serna et al., 2020), and post-processing techniques (Binns, 2018; Caton & Haas, 2024). Most fairness quantification methods focus on categorical-attribute-based biases, i.e., ethnicity, and age (Caton & Haas, 2024). However, geographical bias is in continuous 2D or 3D space, and those methods often fail to account for the intrinsic spatial characteristics of data, such as directional dependence and scale effects.

## 3 PROBLEM SETUP

We first give some formal definitions and mathematical notations that will be used throughout the paper in Section 3.1. Since we will refer to many concepts from classic spatial point pattern analysis, we provide a brief introduction of these concepts in Section 3.2 for the broader AI community.

### 3.1 NOTATIONS AND DEFINITIONS

**Definition 3.1** (Geospatial Dataset). A geospatial dataset $\mathcal{D} := \{(X_i, L_i, y_i)|L_i \in \mathbb{S}^2\}_{i=1}^n$ is a set of triples: $X_i$ is an observation, for example a streetview image; $L_i$ is the geographical location of $X_i$ on the Earth surface $\mathbb{S}^2$, or sometimes approximated by the Euclidean plane $\mathbb{R}^2$; $y_i$ is the task-specific ground-truth for $X_i$, e.g., class labels in classification tasks and real values in regression tasks.

**Definition 3.2** (Model & Predictions). A model $\mathcal{F}$ maps $X_i$ and $L_i$ to a prediction $\hat{y}_i$ of the ground-truth, that is, $\hat{y}_i := \mathcal{F}(X_i, L_i)$.

**Definition 3.3** (Performance Function). A performance function $\boldsymbol{\pi}$ compares the ground-truth $y_i$ and the prediction $\hat{y}_i$ to assign an evaluation $\pi_i := \boldsymbol{\pi}(y_i, \hat{y}_i)$ to a location $L_i$. $\boldsymbol{\pi}$ can be **non-numerical** values. For example, in classification tasks, $\pi_i$ can be binary (correct v.s. incorrect), continuous (logits of predictions), or categorical (human comments).

**Definition 3.4** (Location Map & Performance Map). A *location map* is a set of locations $\mathcal{L}_{\mathcal{D}} := \{L_i\}_{i=1}^n$. A *performance map* is a set of location-evaluation tuples $\mathcal{M}_{\mathcal{D}, \boldsymbol{\pi}} := \{(L_i, \pi_i)\}_{i=1}^n$.

Spatial point patterns always involve multiple locations (a single point will not form "patterns"), so we define the unit to evaluate geo-bias as.

**Definition 3.5** (Region Of Interest (ROI)). A *region of interest (ROI)* $N \in \mathcal{P}(\mathcal{M}_{\mathcal{D}, \boldsymbol{\pi}})$ is a multiple-point subset of the performance map, where $\mathcal{P}(\mathcal{M}_{\mathcal{D}, \boldsymbol{\pi}})$ denotes the power set of $\mathcal{M}_{\mathcal{D}, \boldsymbol{\pi}}$.

**Definition 3.6** (Local Geo-Bias Score). A function $\gamma : \mathcal{P}(\mathcal{M}_{\mathcal{D}, \boldsymbol{\pi}}) \to \mathbb{R}$ measures the strength of geo-bias in an ROI $N$. We call $\gamma(N)$ a *local geo-bias score*.

**Definition 3.7** (Global Geo-Bias Score). Let $\mathcal{N} := \{N_m \in \mathcal{P}(\mathcal{M}_{\mathcal{D}, \boldsymbol{\pi}})\}_{m=1}^M$ be the set of all ROIs where we intend to measure geo-bias. We compute $\gamma(N_m)$ for each $N_m \in \mathcal{N}$ and use their weighted sum as the *global geo-bias score*.

### 3.2 RELATED CONCEPTS FOR SPATIAL POINT PATTERN ANALYSIS

**Unmarked & Marked Point Patterns.** In spatial point pattern analysis, *unmarked* patterns only consider the locations of points, while *marked* patterns consider both the locations and the attribute values of the points. For example, the locations where bat-eared fox is observed is a typical unmarked

spatial point pattern, since it only considers the spatial distributions of species occurrences, while no value is attached to each location. On the other hand, a set of geo-tagged soil samples is a marked spatial point pattern in which we consider both the geolocations of these samples and these samples' soil attribute values (e.g., soil moisture, pH, salinity, etc.). Based on the above definitions, a location map $\mathcal{L}_{\mathcal{D}} := \{L_i\}_{i=1}^n$ is an **unmarked spatial point pattern** while a performance map $\mathcal{M}_{\mathcal{D},\boldsymbol{\pi}} := \{(\pi_i, L_i)\}_{i=1}^n$ is a **marked spatial point pattern**.

**Summary Statistics of Spatial Point Patterns.** In spatial point pattern analysis, various summary statistics are developed to quantify the spatial autocorrelation or spatial heterogeneity of a spatial point pattern. These statistics can be classified into two categories: first-order and second-order summary statistics. The **first-order summary statistics** (O'sullivan, 2003; Ben-Said, 2021) focus on quantifying the variation of the intensity of point patterns (for unmarked point patterns) and the expectation of attribute values (for marked point patterns) across a study area. Examples include nearest neighbor distribution function, spherical contact distribution (Ben-Said, 2021), Moran's I (Moran, 1950), LISA (Anselin, 1995), Geary'C, and Local Geary's C (Anselin, 2019). In contrast, the **second-order statistics** focus on quantifying the strength of interactions between points according to distance. Examples include Ripley's K-function (Ripley, 1977) and L-functions (Besag, 1977).

## 4 METHODS

In this work, we have two major objectives: 1) Proposing a systematic, theory-supported framework to categorize and interpret geo-bias; 2) Designing spatially explicit geo-bias quantification under this theoretical framework. For the former objective, we propose a novel framework of geo-bias interpretation and categorization based on spatial point pattern analysis concepts introduced in Section 3.2. This framework serves our purposes perfectly because (1) it is compatible with most existing geo-bias metrics, and (2) it clearly points out three key factors we need to consider when designing new geo-bias metrics. For the latter objective, we combine the key factors with information theory to quantify geo-bias. It is because under our theoretical framework, geo-bias is in effect the difference between the observed spatial point patterns in the location/performance maps and predefined spatially homogeneous (i.e., "unbiased") spatial point patterns, which can be quantified using information-theoretic terms such as self-information and relative entropy.

In the rest of this section, we will firstly introduce our theoretical framework in Section 4.1 and use concrete examples to demonstrate how existing geo-bias metrics can be integrated into our framework in Section 4.2. Then, we will propose three novel information-theoretic geo-bias metrics based on our framework in Section 4.3, also explaining their spatial implications with illustrations. Finally, we will describe the algorithms for computing the aforementioned geo-bias scores in Section 4.4.

### 4.1 THEORETICAL FRAMEWORK & CATEGORIZATION OF GEO-BIAS

As we have discussed in Section 3.2, both the location map $\mathcal{L}_{\mathcal{D}}$ and the performance map $\mathcal{M}_{\mathcal{D},\boldsymbol{\pi}}$ of a model are spatial point patterns. Intuitively, *the more "random" the location/performance map looks, the less geographically biased it is.* From the theoretical perspective of spatial point pattern analysis, this intuition can be precisely described as comparing a location/performance map with a predefined, spatially homogeneous reference pattern, and the more different they are the more geo-biased the model is (against the chosen reference pattern). The metrics we use to quantify the difference between patterns are naturally valid geo-bias metrics. For example, if we assume that an ideally unbiased model should perform uniformly well across the space with Gaussian fluctuations, we can compute the Kolmogorov–Smirnov (KS) test statistics of the performance map against a Gaussian distribution and use it as a geo-bias metric – that is, the larger the statistics, the less likely the model performs uniformly well as we hypothesized, thus more geo-biased under our assumption of homogeneity.

Based on this interpretation, we can summarize three key factors that differentiate one geo-bias metric from another: (1) the **map** (location map or performance map) we use for comparison, (2) the **reference pattern** (i.e., the desired unbiased pattern), and (3) the **difference measure** between the map and the reference pattern. These factors enable a neat categorization of geo-bias: we can categorize a geo-bias metric as (1) "Unmarked" v.s. "Marked" based on which map it uses, (2) "Gaussian", "Poisson", "Permutation", etc., based on which reference pattern it uses, and (3) "Statistical", "Information-Theoretic", etc., based on which difference measure it uses.

Another important dimension of geo-bias categorization is **"First-Order" v.s. "Second-Order"**. As discussed in Section 3.2, a geo-bias metric that summarizes the heterogeneity of a spatial point pattern can be either first-order if it is an averaged number over all points, or second-order if it is a

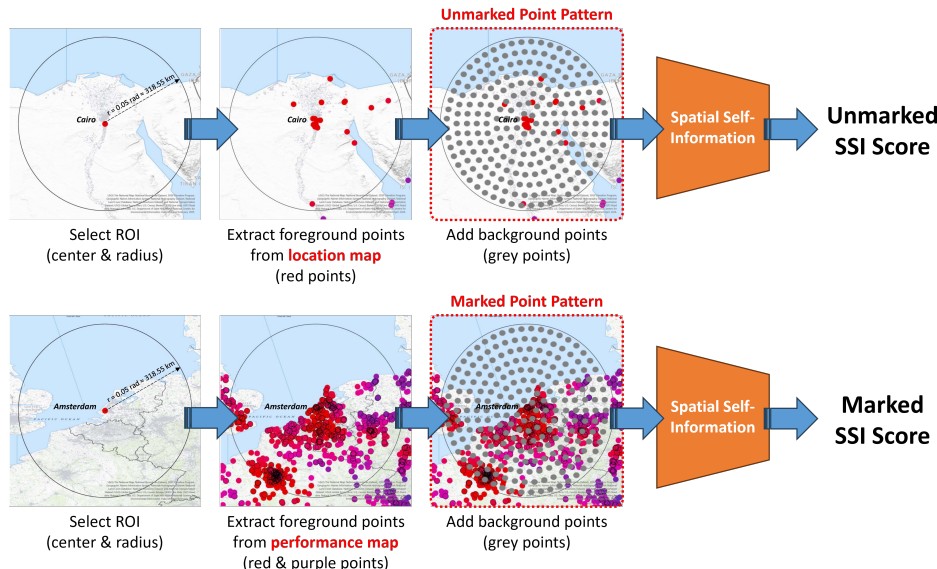

Figure 2: Workflows of Unmarked SSI and Marked SSI computation.

function of point interactions such as covariances against distance. Most existing geo-bias metrics (Manvi et al., 2024; Xie et al., 2022; Wu et al., 2024), as well as the novel geo-bias scores we propose in this paper, are first-order. We leave the investigation of second-order geo-bias metrics to the future.

## 4.2 EXISTING GEO-BIAS METRICS: SPATIAL SELF-INFORMATION (SSI) SCORES

We use two recently proposed but commonly used geo-bias metrics as concrete examples to demonstrate the applicability of our framework discussed in Section 4.1. **Unmarked SSI Score.** Unmarked SSI Score is proposed in (Wu et al., 2024) as a measure of dataset geo-bias, i.e., whether the data points are uniformly distributed across the space. Figure 2 illustrates the computation workflow. According to our theoretical framework, this metric is (1) "Unmarked" because the location map is used, (2) "Permutation" because the reference pattern is a random permutation of foreground/background points, and (3) "Information-Theoretic" because the difference measure is the self-information of the location map (implicitly against the reference pattern) Wang et al. (2024). **Marked SSI Score.** Marked SSI Score is proposed in (Wu et al., 2024) as a measure of model performance geo-bias, i.e., whether the model performance (accuracy, MSE, etc.) is consistently good across the space. Figure 2 illustrates the computation workflow. Similarly, this metric is (1) "Marked" because the performance map is used, (2) "Permutation" because the reference pattern is a random permutation of good/bad performance points, and (3) "Information-Theoretic" because the difference measure is the self-information of the performance map (implicitly against the reference pattern).

## 4.3 NOVEL GEO-BIAS METRICS: SPATIAL RELATIVE-ENTROPY (SRE) SCORES

We have demonstrated the power of our theoretical framework in decomposing a geo-bias metric into three dimensions: the map, the reference pattern, and the difference measure. This decomposition also helps us be more purposeful when designing geo-bias metrics. We notice that for Unmarked SSI and Marked SSI, the difference measure, i.e., self-information (also known as "surprisal" in information theory), effectively accounts for the geo-bias specifically related to *spatial proximity*. This is because Wang et al. (2024) proves that the self-information of a spatial point pattern is an alternative quantification of Moran's I (Moran, 1950), which measures the autocorrelation among spatial neighbors. This finding inspires us: what if we want to measure the geo-bias related to other important spatial factors?

### 4.3.1 SPATIAL MOTIVATIONS OF SRE SCORES

Analogous to the famous Simpson's Paradox in statistics (Pearl, 2022), spatial heterogeneity makes the conclusions drawn from spatial data sensitive to the way we partition the space (Mai et al., 2025). It is also related to the fundamental Modifiable Unit Area Problem (Openshaw, 1984; Fotheringham & Wong, 1991; Goodchild, 2022; Chen et al., 2022), and recognized as a major source of bias in various domains such as ecology (Jelinski & Wu, 1996; Swift et al., 2008) and urban geography

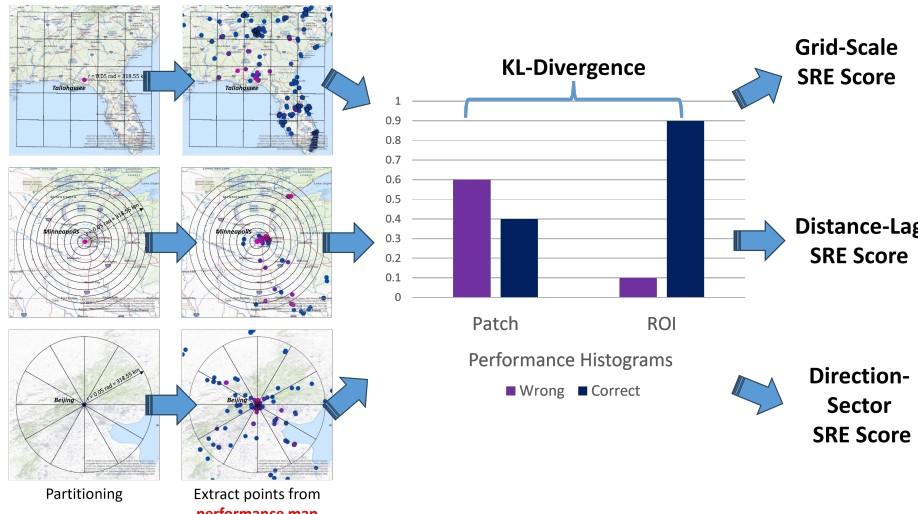

Figure 3: Workflows of SRE Scores computation.

(Deng et al., 2024). For example, if a model is sensitive to directions, it will demonstrate strong directional heterogeneity, i.e., the model performs significantly differently on data points drawn from different directions. Figure 1 clearly illustrates that the model performance in each disjoint area (e.g. square, ring, sector) differs from the overall performance, indicating that there is geo-bias.

In this paper, we are interested in (but not limited to) three specific partitionings: *scale-grid*, *distance-lag*, and *direction-sector*. We thus design three novel, model-agnostic and spatially explicit geo-bias metrics according to the partitionings called *Scale-Grid SRE Score*, *Distance-Lag SRE Score*, and *Direction-Sector SRE Score*. They correspond to three important spatial factors – *multi-scalability*, *distance decay*, and *anisotropy*, respectively. SRE stands for **S**patial **R**elative-**E**ntropy, because the difference measures we use in these metrics is the Kullback–Leibler (KL) divergence, also known as *relative entropy*.

### 4.3.2 FORMAL DEFINITIONS OF SRE SCORES

**Definition 4.1** (Partition Function, Partitioning & Patch). $\mathbb{PAR}$ is called a *partition function* if it divides the spatial area $A$ of an ROI $N$ into a set of disjoint sub-areas $\mathcal{A} := \bigcup A_k$ and maps $N$ into disjoint subsets $P_k := \{(L_i, \pi_i)|L_i \in A_k\}$. The set of disjoint subsets $\Pi := \bigcup P_k$ is called a *partitioning* of $N$, and each subset $P_k$ is called a *patch* in the partitioning.

**Definition 4.2** (ROI & Patch Performance Distribution). Let $h$ be a mapping from a set of location-evaluation tuples to a probability distribution, e.g., the normalized histogram of correct and wrong predictions. $h(N)$ and $h(P_k)$ are called an *ROI performance distribution* and a *patch performance distribution*, respectively.

**Definition 4.3** (Local SRE Score). Let $d$ be a difference measure between distribution $h(P_k)$ and distribution $h(N)$, e.g. Kullback–Leibler divergence. The *Local SRE Score* of ROI $N$ is defined as $\gamma_{\text{SRE}}(N) := \#P_k/\#N \sum_k d(h(P_k), h(N))$.

**Definition 4.4** (Global SRE Score). The *Global SRE Score* is defined as the weighted sum of all Local SRE Scores: $\Gamma_{\text{SRE}} := \sum_m w_m \gamma_{\text{SRE}}(N_m)$. $w_m$ is the user-defined weight for ROI $N_m$.

The partition functions used in this paper include: (1) **Scale-Grid:** Partition an ROI into equal-size squares; (2) **Distance-Lag:** Partition an ROI into equal-width concentric rings. (3) **Direction-Sector:** Partition an ROI into equal-angle sectors. However, we encourage researchers to design their own partitioning based on their needs and domain knowledge and enlarge the family of SRE Scores.

### 4.4 IMPLEMENTATIONS OF SSI AND SRE GEO-BIAS SCORES IN GEOBS

Finally, we will discuss the detailed implementation of both SSI Scores and SRE Scores computation in our GeoBSPython package.

For SSI Scores, the formal definitions and theoretical formulas can be found in Wu et al. (2024) and Wang et al. (2024). While the SSI Score algorithm we use (Algorithm 2 in Appendix B.1) remains mostly unaltered, the quality and stability of implementation in our GeoBS package are significantly

---

**Algorithm 1** Local SRE Algorithm

---

**Input** : Performance map $\mathcal{M}_{\mathcal{D},\boldsymbol{\pi}} := \{(\pi_i, L_i)\}_{i=1}^n$. Location of the ROI's center point $L_c$. Radius of the ROI $r$. Distance function $d_c$. Partition algorithm $\mathbb{PAR}$ (Scale-Grid Partition, Distance-Lag Partition, Direction-Sector Partition). Histogram bins $(b_0, b_1, \cdots, b_H)$. KL divergence function $D_{\text{KL}}$.

**Output** : A local SRE score $\gamma_{\text{SRE}}$ for the ROI centered at $L_c$ with radius $r$ and partition function $\mathbb{PAR}(N)$.

1 Retrieve points in ROI: $N \leftarrow \{(\pi_i, L_i) \mid d_c(L_i, L_c) < r\}$;

2 Partition ROI: $\Pi \leftarrow \mathbb{PAR}(N)$;

3 Compute ROI histogram: $h(N) \leftarrow \{\#\{b_j \leq \pi_s < b_{j+1}\} \mid (\pi_s, L_s) \in N\}$;

4 Initialize $\gamma_{\text{SRE}} \leftarrow 0$;

5 For $P_k \in \Pi$:

  Compute patch histogram: $h(P_k) \leftarrow \{\#\{b_j \leq \pi_t < b_{j+1}\} \mid (\pi_t, L_t) \in P_k\}$;

  Compute KL divergence: $d(P_k) \leftarrow D_{\text{KL}}(h(N) \,\|\, h(P_k))$;

  Accumulate weighted divergence: $\gamma_{\text{SRE}} \leftarrow \gamma_{\text{SRE}} + \frac{\#P_k}{\#N} \cdot d(P_k)$;

6 **return** $\gamma_{\text{SRE}}$

---

improved over the original PyGBS package (Wu et al., 2024). The most important changes include: (1) we modularized the SSI Score algorithm so that it shares common data preprocessing and postprocessing procedures with our SRE Scores, which saves up to 20% of computational costs since these procedures only need to run once and work for all five geo-bias scores; (2) we solved the `nan` issues commonly encountered in the original implementation by introducing a background point generator which automatically adjusts the point density $\rho$ to avoid "divided by zero" errors; (3) we implement the Fibonacci Lattice algorithm to generate background points in place of the original random background point generation, which solved the reproducibility issue of the original implementation (i.e., two runs of SSI Scores may differ due to different random background points).

For SRE Scores, we choose to use the KL divergence $D_{\text{KL}}(h(N), h(P_k))$ as $d$ in Definition 4.3. The choice of KL divergence is based on two considerations: (1) $d$ needs to be a difference measure between probability distributions, among which KL divergence is the most commonly used and most generally applicable (e.g., KL divergence can be computed in $\mathcal{O}(n)$ complexity for any discrete distributions while Wasserstein distance may be as complex as $\mathcal{O}(n^3)$ (Edmonds & Karp, 1972)); (2) KL divergence has physical meanings in that it measures the information gap between two distributions (i.e., *relative*-entropy), which can be interpreted as the bits needed to transform a geo-biased map into an unbiased one, potentially useful in transfer learning. We also choose to use the normalized ROI size $\#N_m / \sum_m \#N_m$ as the weight $w_m$, but one can always use other factors of interest, such as area, population, and GDP for weighting.

The computation of Local SRE Scores is described in Algorithm 1. Intuitively, if the model performance does not have geo-bias under the given partitioning $\mathbb{PAR}$, the probability of encountering good/bad performance points in each patch $P_k$ (e.g., square/ring/sector) should be similar. We use the histograms $h(N) \leftarrow \{\#\{b_i \leq \pi_s < b_{i+1}\} \mid (\pi_s, L_s) \in N\}$ and $h(P_k) \leftarrow \{\#\{b_i \leq \pi_t < b_{i+1}\} \mid (\pi_t, L_t) \in P_k\}$ as the empirical distributions of the model performance in the corresponding ROI $N$ and Patch $P_k$. Then, the KL-divergence between $h(N)$ and $h(P_k)$ is used as the measure of information gap between the patch and the entire ROI. Finally, the weighted sum (over the sizes of patches) of all KL-divergences across the ROI $N$ is used as the SRE Score for the local region of interest (see Definition 4.3).

## 5 EXPERIMENTS

The experiments in this research consist of three tasks:

**1. Geo-Aware Image Classification:** We conduct image classification on three geo-tagged image datasets: iNat2017, iNat2018, and fMoW. The models evaluated include both an image-only classifier (No Prior) and image classifiers enhanced with four commonly used location encoders: Radial Basis Function (RBF), Space2Vec-theory (Space2Vec), NeRF, and Sphere2Vec-sphereC (Sphere2Vec). The model performance is measured in binary numbers: 0 for wrong classes and 1 for correct classes.

**2. Geo-Aware Image Regression:** We use the same models as in general image classification tasks, except that we predict continuous values which represent population density, forest coverage percentage, nightlights luminosity, and other indices at the given location. The benchmark we use is MOSAIKS (Rolf et al., 2021). The model performance is measured in binary numbers: 0 for absolute errors smaller than the empirical variance of all prediction errors and 1 otherwise.

| Abbreviation | U-SSI | M-SSI | SG-SRE | DL-SRE | DS-SRE | SPAD |
|---|---|---|---|---|---|---|
| Meaning | Unmarked SSI | Marked SSI | Scale-Grid SRE | Distance-Lag SRE | Direction-Sector SRE | SPace-As-Distribution Score |

Table 1: Abbreviations used in experiments.

**3. Remote Sensing (RS) Image Classification:** We experiment with four RS image classification datasets: EuroSat (Helber et al., 2019), fMoW-sentinel (Cong et al., 2022), WorldStrat-IPCC, and WorldStrat-LCCS (Cornebise et al., 2022). To study the geo-bias of different FMs, we pick 2 remote sensing **foundation models**, e.g., CROMA (Fuller et al., 2024) and SATMAE (Cong et al., 2022), along with an LLM, GPT-4o, to classify multi-spectral satellite imagery. The model performance is measured in binary numbers: 0 for wrong classes and 1 for correct classes. We report the accuracy for classification and $R^2$ for regression. We also report all five global geo-bias scores

Table 2: Accuracy and Global Geo-Bias Scores of geo-tagged image classification. All geo-bias scores use an ROI radius of 0.05 radian. **Bold** numbers indicate the best performance or the lowest geo-bias.

| | Model | Acc ↑ | U-SSI ↓ | M-SSI ↓ | SG-SRE ↓ | DL-SRE ↓ | DS-SRE ↓ | SPAD ↓ |
|---|---|---|---|---|---|---|---|---|
| fMoW | **Hyperparam** | - | - | - | 0.01 | 0.005 | 12 | - |
| | No Prior | 69.83 | 546.59 | **432.19** | 13.94 | 6.70 | 7.68 | **18.20** |
| | rbf | 70.64 | **545.82** | 436.38 | 14.67 | 6.76 | 7.80 | 18.98 |
| | Space2Vec | 70.49 | 546.22 | 436.46 | 14.29 | 6.82 | 8.15 | 18.75 |
| | NeRF | 69.92 | 546.55 | 433.56 | **13.87** | **6.66** | **7.61** | 18.56 |
| | Sphere2Vec | **70.66** | 546.85 | 436.88 | 14.64 | 6.81 | 7.84 | 18.76 |
| iNat2017 | **Hyperparam** | - | - | - | 0.005 | 0.005 | 8 | - |
| | No Prior | 63.27 | 552.87 | **247.84** | 14.16 | 3.54 | 2.78 | 19.65 |
| | rbf | 68.29 | **552.46** | 289.08 | **13.35** | **3.29** | **2.73** | 20.02 |
| | Space2Vec | 68.30 | 556.42 | 289.45 | 13.98 | 3.58 | 2.85 | 19.31 |
| | NeRF | 68.68 | 554.83 | 292.63 | 14.14 | 3.48 | 2.80 | 19.14 |
| | Sphere2Vec | **69.16** | 555.67 | 297.54 | 13.79 | 3.51 | 2.83 | **18.38** |
| iNat2018 | **Hyperparam** | - | - | - | 0.025 | 0.005 | 8 | - |
| | No Prior | 60.20 | **447.25** | **170.95** | 2.12 | 1.95 | 1.41 | 21.84 |
| | rbf | 63.89 | 462.59 | 185.43 | 2.14 | 1.99 | 1.39 | 21.92 |
| | Space2Vec | **73.52** | 460.97 | 254.73 | **1.67** | **1.47** | **1.20** | 18.88 |
| | NeRF | 72.91 | 458.90 | 248.31 | 1.69 | 1.48 | 1.21 | **18.69** |
| | Sphere2Vec | 72.93 | 459.57 | 251.40 | 1.80 | 1.56 | 1.27 | 18.73 |

(Unmarked SSI, Marked SSI, Scale-Grid SRE, Distance-Lag SRE, and Direction-Sector SRE) together with a baseline *SPace-As-Distribution Score* proposed by Xie et al. (2022). The global scores are computed over ROIs that **are not all 0s or all 1s** for the sake of computational stability (in this case, the scores could become infinity). The abbreviations we use throughout the experiment section are listed in Table 1. For more information about the experiment setup, please see Appendix Table 5.

The hyperparameters used in the experiments are: radius of ROI, grid size for SG-SRE, lag width for DL-SRE, the number of splits for DS-SRE. The choice of hyperparameters affects the amount of data points we use in computing geo-bias scores. For example, if the radius of an ROI is 1 km, it is likely that each ROI only contains one data point, and we are unable to compute geo-bias scores. In order to avoid such extreme cases, we select the appropriate hyperparameters for each dataset based on their data spatial distribution. The principle is: (1) each ROI contains at least 100 points, and (2) at least 2 patches in one ROI contain more than 10 points. All hyperparameters are reported in the experiment tables. As to the baseline SPAD Score, it ranges from 0 to 100 and is calculated using a maximum of

Table 3: $R^2$ and Global Geo-Bias Scores of geo-aware neural regression. All experiments use ROI radius 0.2 radian, scale 0.1 radian, lag 0.05 radian, number of splits 8. **Bold** numbers indicate the best performance or the lowest geo-bias.

| | Model | $R^2$ ↑ | U-SSI ↓ | M-SSI ↓ | SG-SRE ↓ | DL-SRE ↓ | DS-SRE ↓ | SPAD ↓ |
|---|---|---|---|---|---|---|---|---|
| Population Density | No Prior | 0.38 | **13.11** | 4.26 | 5.58 | 28.75 | 22.44 | 21.68 |
| | rbf | 0.25 | 14.13 | 3.97 | **0.47** | **22.33** | 18.29 | 21.34 |
| | Space2Vec | 0.57 | 15.53 | 3.04 | 1.65 | 33.53 | 22.24 | **20.68** |
| | NeRF | 0.60 | 17.51 | 3.38 | 18.71 | 26.86 | 16.47 | 22.15 |
| | Sphere2Vec | **0.63** | 15.01 | **2.82** | 4.57 | 25.91 | **14.86** | 21.90 |
| Forest Cover | No Prior | 0.52 | 20.32 | 3.51 | 47.78 | 144.54 | 297.32 | **23.43** |
| | rbf | 0.54 | 19.60 | **2.63** | 37.28 | **126.78** | 299.58 | 24.80 |
| | Space2Vec | **0.73** | 19.48 | 3.87 | 50.67 | 164.10 | 343.70 | 25.15 |
| | NeRF | 0.68 | **18.08** | 2.85 | 37.98 | 149.73 | 305.72 | 25.84 |
| | Sphere2Vec | **0.73** | 21.53 | 4.40 | **21.23** | 130.51 | **284.26** | 24.95 |
| Nightlight Luminosity | No Prior | 0.33 | 20.48 | 2.51 | 7.45 | 47.32 | 19.26 | 23.25 |
| | rbf | 0.32 | 21.62 | 3.71 | 25.40 | 21.81 | 51.73 | 22.92 |
| | Space2Vec | 0.21 | 20.19 | 2.96 | **4.11** | **7.65** | **12.05** | **21.57** |
| | NeRF | 0.23 | 20.67 | 2.99 | 9.19 | 17.71 | 78.18 | 22.59 |
| | Sphere2Vec | **0.35** | **20.13** | **2.18** | 10.23 | 9.76 | 40.05 | 21.60 |
| Elevation | No Prior | 0.27 | 22.33 | 3.93 | 30.71 | 21.76 | 106.08 | 21.26 |
| | rbf | 0.39 | 21.79 | 4.50 | 9.61 | 6.97 | 24.96 | **19.62** |
| | Space2Vec | 0.78 | **20.22** | **3.51** | 4.29 | **6.92** | **16.60** | 20.74 |
| | NeRF | 0.76 | 20.82 | 4.25 | 7.43 | 9.44 | 26.37 | 19.88 |
| | Sphere2Vec | **0.82** | 21.25 | 4.84 | 4.42 | 17.51 | 26.27 | 20.44 |

100 rows, 100 columns, and a partitioning sample size of 100 (Xie et al., 2022). For ablation studies on the influence of hyperparameters, please see Table 6 and Table 7 in the Appendix.

## 5.1 GEO-BIAS OF TASK-SPECIFIC GEOAI MODELS

Table 2 reports the geo-tagged image classification geo-bias. By comparing the model accuracy with the geo-bias scores, we can see that there is no strong correlation, which means geo-bias scores are a (relatively) independent dimension of evaluation, and it is not sufficient to only report the overall performance. Moreover, we observe that the geo-bias of task-specific GeoAI models tends to be mostly dependent on datasets rather than on models. For example, all models have significantly lower

SRE Scores on the iNat2018 dataset (notice that SSI and SRE Scores are log-based), which suggests that iNat2018 might have more spatially balanced data.

In contrast, while the NeRF model shows very low geo-bias in terms of scale, distance, and direction on the iNat2017 and iNat2018 datasets, it performs significantly more biased on the fMoW dataset. Similar observations can be made from Table 3 which reports the geo-aware image regression geo-bias, where the Forest Cover dataset shows drastically larger geo-bias in terms of all three SRE Scores. See Figure 4 in Appendix B.2 for an intuitive visualization.

Our hypothesis is that it is because such GeoAI models explicitly leverage the geographical metadata (e.g., latitudes and longitudes) for predictions, which causes the model to overfit to the spatial distributions of training data (Mai et al., 2020; 2023). If the dataset is geo-biased in data sampling, the performance will also suffer regardless of which model is used. In this case, we should focus on improving the data quality, such as class balance, spatial coverage, etc.

## 5.2 GEO-BIAS OF REMOTE SENSING FOUNDATION MODELS

Foundation models, which are trained on massive data, are believed to suffer less from data bias. This partially matches our observation. Table 4 reports the performance of ChatGPT and two remote sensing foundation models with variations (`ft` stands for "finetuning" and `lp` stands for "linear probing"). The geo-bias scores are significantly lower than the task-specific counterparts. Besides, unlike the task-specific case, the differences in geo-bias scores of the same model across different datasets are not prominent. Instead, we observe that while the CROMA ft model outperforms SatMAE almost consistently, it also has way stronger geo-bias of all types. See Figure 5 in Appendix B.2 for an illustration of this phenomenon.

Our hypothesis is that, while foundation models are trained on a massive amount of data and thus less affected

Table 4: Accuracy and Global Geo-Bias Scores of remote sensing image classification. All geo-bias scores use an ROI radius of 0.01 radian. **Bold** numbers indicate the best performance or the lowest geo-bias. **Bold** numbers indicate the best performance or the lowest geo-bias scores.

| | Model | Acc↑ | U-SSI↓ | M-SSI↓ | SG-SRE↓ | DL-SRE↓ | DS-SRE↓ | SPAD↓ |
|---|---|---|---|---|---|---|---|---|
| **fMoW-sentinel** | **Hyperparam** | - | - | - | 0.01 | 0.01 | 8 | - |
| | GPT-4o | 5.72 | **516.80** | **63.96** | 3.81 | 1.32 | 0.76 | 18.25 |
| | CROMA ft | 52.67 | 560.80 | 447.89 | 61.47 | 16.79 | 19.94 | 39.19 |
| | CROMA lp | 31.46 | 560.11 | 466.31 | 152.72 | 38.69 | 42.17 | 36.37 |
| | SatMAE ft | **64.77** | 560.96 | 16.29 | **2.16** | **0.57** | **0.74** | 12.84 |
| | SatMAE lp | 62.76 | 561.29 | 14.06 | 2.47 | 0.61 | 0.88 | **12.36** |
| **WorldStrat-LCCS** | **Hyperparam** | - | - | - | 0.05 | 0.005 | 8 | - |
| | GPT-4o | 41.33 | 399.27 | 276.18 | 7.87 | 54.87 | 62.21 | 66.93 |
| | CROMA ft | **60.78** | **354.01** | 275.65 | 12.91 | 18.89 | 23.46 | 63.23 |
| | CROMA lp | 58.73 | 369.52 | 305.35 | 3.98 | 10.59 | 21.51 | 66.56 |
| | SatMAE ft | 52.37 | 418.63 | **6.00** | **0.06** | **0.11** | **0.14** | 16.51 |
| | SatMAE lp | 44.29 | 416.44 | 6.95 | **0.06** | 0.12 | 0.16 | **15.29** |
| **WorldStrat-IPCC** | **Hyperparam** | - | - | - | 0.05 | 0.005 | 8 | - |
| | GPT-4o | 51.92 | 404.86 | 200.12 | 3.82 | 12.06 | 12.40 | 56.40 |
| | CROMA ft | **69.61** | **359.10** | 251.67 | 7.64 | 13.21 | 14.67 | 52.27 |
| | CROMA lp | 65.79 | 379.79 | 271.37 | 4.64 | 8.28 | 9.09 | 56.12 |
| | SatMAE ft | 66.56 | 410.33 | 19.23 | **0.07** | 0.18 | **0.15** | **15.81** |
| | SatMAE lp | 45.36 | 416.16 | **7.06** | 0.08 | **0.14** | 0.16 | 16.07 |
| **EuroSAT** | **Hyperparam** | - | - | - | 0.01 | 0.005 | 12 | - |
| | GPT-4o | 44.89 | 119.43 | 79.59 | 2.62 | 1.29 | 0.64 | 53.52 |
| | CROMA ft | **97.43** | 115.72 | 96.58 | 0.25 | 0.67 | 0.48 | **8.67** |
| | CROMA lp | 92.87 | **100.00** | 60.35 | 0.56 | 0.44 | 0.37 | 19.23 |
| | SatMAE ft | 74.30 | 115.93 | 13.02 | 0.03 | **0.07** | 0.05 | 15.65 |
| | SatMAE lp | 56.54 | 113.19 | **6.43** | **0.02** | 0.07 | 0.06 | 34.91 |

by data geo-bias, their powerful learning capability may overfit the implicit geographical information in the data, especially considering that the EuroSAT dataset only covers Europe, and the implicit spatial patterns can be easily acquired without explicit input of geolocations. In other words, whether a foundation model is prone to geo-bias might be more dependent on its own model architecture, i.e., how suitable this model is for learning spatial features.

## 6 CONCLUSION, LIMITATION & FUTURE WORK

Our work is an example of using the rich domain knowledge (spatial data analysis, point pattern analysis) to guide AI research. We provide a powerful framework of designing geo-bias scores that explicitly describes what spatial factors you care about and gives clear-cut, information-theoretic interpretations of the evaluation results. We see this as a great opportunity to encourage researchers to report geo-bias scores in their work so that we are not only racing for higher model performance, but also keeping in mind the spatial fairness issues behind it. In this paper, we limit our discussion on first-order, relative-entropy-based geo-bias scores, but we will design more geo-bias scores in our future work that deal with other intricate spatial factors, for example, network and time-space. Besides, since the geo-bias scores we propose are based on self-information and relative information, they are differentiable and compatible with most existing training objectives. We see great potential in introducing the geo-bias scores as debiasing loss functions and help train more fair models.

## REPRODUCIBILITY STATEMENT

We provide detailed descriptions of our methods, algorithm implementation, models, datasets and hyperparameters in Section 3, 4, 5 and Appendix B. To support replication, we have uploaded anonymized source code as supplementary materials.

## ETHICS STATEMENT

We use only publicly available datasets and established benchmarks for evaluation; experiments operate at regional/task level rather than individual profiling. All datasets are used under their licenses, and results are reported for scientific benchmarking only. We adhere to the ICLR Code of Ethics throughout submission, review, and discussion.

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

# A  APPENDIX

# B  APPENDIX

## B.1  Supplementary Information of SSI Scores

Below is the algorithm we use to implement the SSI Scores described in (Wu et al., 2024). Notice that it is slightly different from the original algorithm in the way of generating background points.

---

**Algorithm 2** Local Unmarked/Marked SSI Algorithm

---

**Input**  : Performance map $\mathcal{M}_{\mathcal{D},\boldsymbol{\pi}} := \{(\pi_i, L_i)\}_{i=1}^n$. Location of the ROI's center point $L_c$. Radius of the ROI $r$. Great circle distance $d_c$. Background point density $\rho$. Moran's I conversion algorithm $\mathbb{SSI}$ (Wang et al., 2024).

**Output** : A local Unmarked/Marked SSI score $\gamma_{\text{SSI}}$ for the ROI centered at $L_c$ with radius $r$.

7  Retrieve the points within the ROI.

   For Unmarked SSI: $N \leftarrow \{(1, L_i) \in \mathbb{S}^2 | (\pi_i, L_i) \in \mathcal{M}_{\mathcal{D},\boldsymbol{\pi}}, d_c(L_i, L_c) < r\}$;

   For Marked SSI: $N \leftarrow \{(\pi_i, L_i) \in \mathbb{S}^2 | (\pi_i, L_i) \in \mathcal{M}_{\mathcal{D},\boldsymbol{\pi}}, d_c(L_i, L_c) < r\}$;

8  Use the Fibonacci Lattice method to generate $\rho\pi r^2$ evenly distributed background points within the ROI:

   $B \leftarrow \{(0, L_j) \in \mathbb{S}^2 | d_c(L_j, L_c) < r\}$;

9  Merge $N$ and $B$: $M \leftarrow N \bigcup B$;

10  Compute the local Unmarked/Marked SSI score: $\gamma_{\text{SSI}} \leftarrow \mathbb{SSI}(M)$

11  **return** $\gamma_{\text{SSI}}$

---

## B.2  Supplementary Figures

We visualize the spatial distributions of reported geo-bias scores on selected datasets and tasks. In all visualizations, the values are normalized to the range of 0 to 1. A darker red color indicates higher bias/error. The visual illustration conforms with the conclusions made in Section 5.

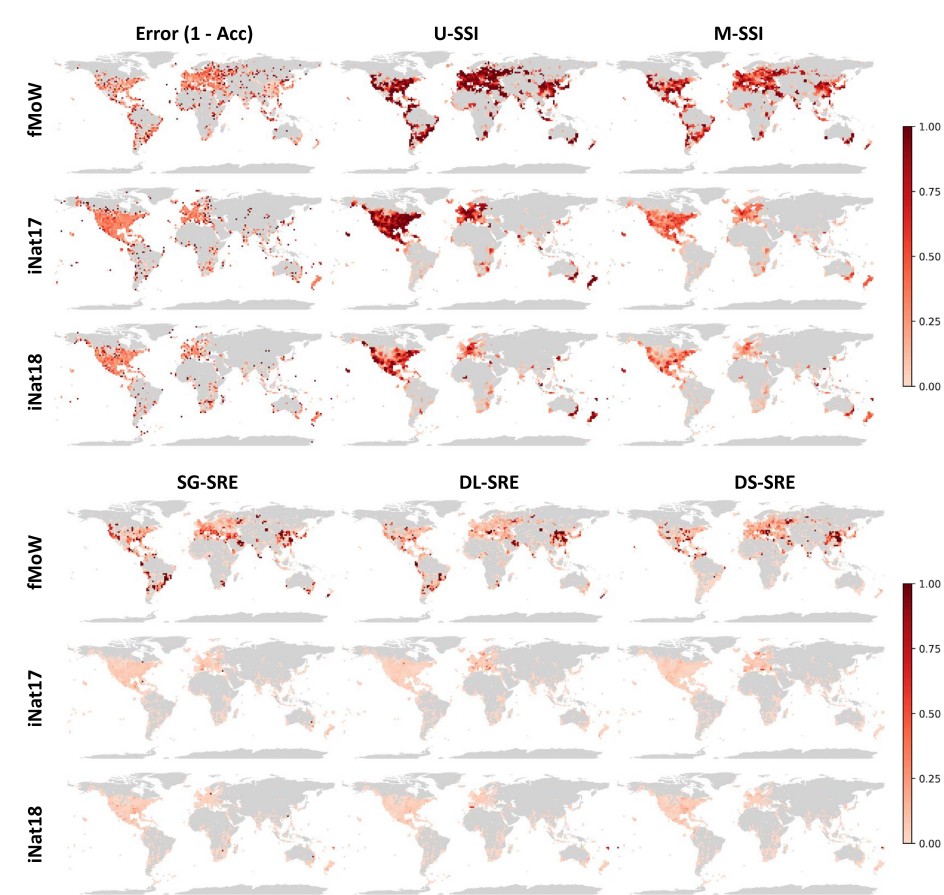

Figure 4: Geographical distributions of error rate and local geo-bias scores of NeRF on fMoW, iNaturalist2017 and iNaturalist2018 on different datasets.

## B.3 SUPPLEMENTARY TABLES

### B.3.1 EXPERIMENT SETUP

### B.3.2 HYPERPARAMETER SENSITIVITY OF SSI SCORES

### B.3.3 HYPERPARAMETER SENSITIVITY OF SRE SCORES

| Dataset | Description |
|---|---|
| iNat2017 | A global species recognition dataset designed for the iNaturalist 2017 challenges (iNaturalist 2018 competition dataset), containing 675,170 images and 5,089 unique categories. |
| iNat2018 | A global species recognition dataset designed for the iNaturalist 2018 challenges (iNaturalist 2018 competition dataset), containing 461,939 images and 8,142 unique categories. |
| fMoW | A global RS image classification dataset (Christie et al., 2018) that includes RS images representing a wide range of land use types. |
| fMoW-sentinel | A global Sentinel-2 dataset cross-referenced with fMoW, as a benchmark for training models on multi-spectral satellite imagery. (Cong et al., 2022) |
| WorldStrat-LCCS | A global collection of high-resolution satellite imagery using the Land Cover Classification System (LCCS) to categorize land cover types. (Cornebise et al., 2022) |
| WorldStrat-IPCC | A global collection of high-resolution satellite imagery using the Intergovernmental Panel on Climate Change (IPCC) classification system. (Cornebise et al., 2022) |
| EuroSAT | A European dataset designed for land use and land cover classification using satellite imagery. Helber et al. (2019) |
| Population Density | A uniformly-at-random distributed global RS image dataset Wu et al. (2024) that contains 425,637 samples and corresponding estimations of population density. |
| Forest Cover | A uniformly-at-random distributed global RS image dataset Wu et al. (2024) that contains 498,106 samples and corresponding estimations of forest cover rate. |
| Nightlights Luminosity | A uniformly-at-random distributed global RS image dataset Wu et al. (2024) that contains 492,226 samples and corresponding nightlights luminosity. |
| Elevation | A uniformly-at-random distributed global RS image dataset Wu et al. (2024) that contains 498,115 samples and corresponding elevation. |
| **Model** | **Description** |
| No Prior | Model using image classifier only |
| rbf | Mai et al. (2020) is a kernel-based location encoder |
| Space2Vec (theory) | A multi-scale location encoder for Euclidean space Mai et al. (2020) |
| NeRF | A location encoder using Neural Radiance Fields (NeRF). Mildenhall et al. (2021) |
| Sphere2Vec (sphereC) | A multi-scale location encoder for spherical surface. |
| GPT-4o | A LLM developed by OpenAI |
| CROMA | A RS foundation model with contrastive radar-optical masked autoencoders. (Fuller et al., 2024) |
| SATMAE | A RS foundation model using pre-training transformers for temporal and multi-spectral satellite imagery. (Cong et al., 2022) |

Table 5: Detailed information of datasets and models.

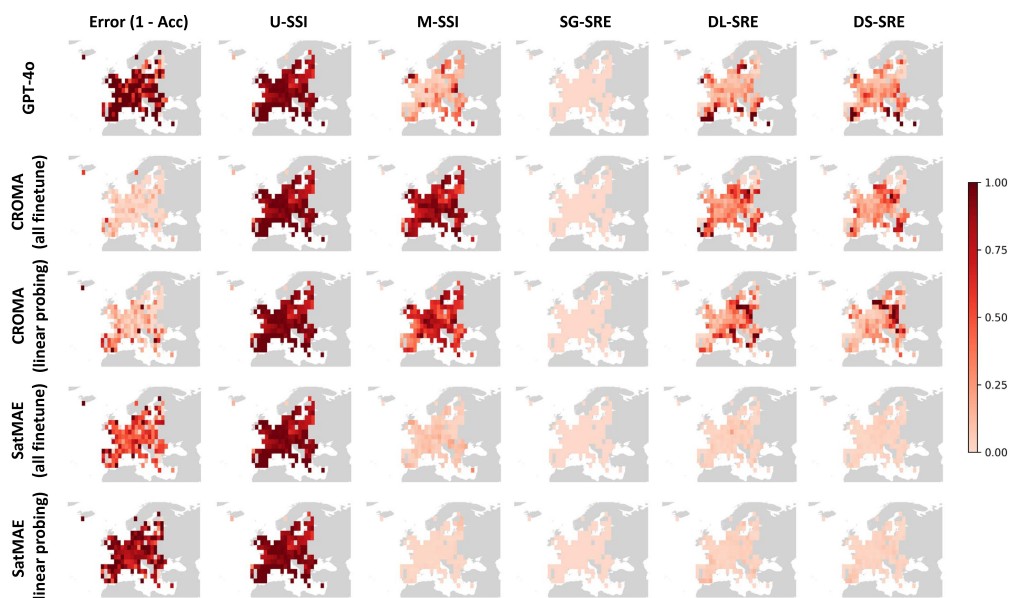

Figure 5: Geographical distributions of error rate and local geo-bias scores of different remote sensing foundation models on EuroSAT. The spatial distributions of U-SSI across models are the same because local U-SSI scores are unmarked and only dependent on data instead of models.

Table 6: Parameter sensitivity test of SSI on iNat2018 dataset. For Unmarked and Marked SSIs, the results with a radius of 0.05, 0.10, 0.15, and 0.20 radians are listed. Bold numbers indicate the chosen parameters.

| Model | U-SSI ↓ | | | | M-SSI ↓ | | | |
|---|---|---|---|---|---|---|---|---|
| | 0.05 | 0.10 | 0.15 | 0.20 | 0.05 | 0.10 | 0.15 | 0.20 |
| rbf | **462.59** | 531.68 | 558.54 | 571.37 | **185.43** | 219.72 | 236.45 | 245.35 |
| Space2Vec-theory | **460.97** | 530.48 | 558.48 | 571.40 | **254.73** | 301.66 | 324.52 | 337.36 |
| NeRF | **458.90** | 529.38 | 557.77 | 570.68 | **248.31** | 294.85 | 317.66 | 330.49 |
| Sphere2Vec-sphereC | **459.57** | 529.49 | 557.94 | 571.04 | **251.40** | 297.71 | 320.62 | 333.46 |

Table 7: Parameter sensitivity test of SRE on NeRF. For Scale-Grid SRE and Distance-Lag SRE, the scales are 0.005, 0.01, and 0.025 radians, respectively. For Direction-Sector SRE, the numbers of splits are 4, 8, and 12. Bold numbers indicate the chosen parameters.

| Dataset | SG-SRE ↓ | | | DL-SRE ↓ | | | DS-SRE ↓ | | |
|---|---|---|---|---|---|---|---|---|---|
| | 0.005 | 0.01 | 0.025 | 0.005 | 0.01 | 0.025 | 4 | 8 | 12 |
| fMoW | 20.26 | **13.87** | 6.98 | **6.66** | 3.36 | 0.99 | 2.77 | 5.50 | **7.61** |
| iNat2017 | **14.14** | 10.50 | 4.87 | **3.48** | 2.08 | 0.78 | 1.54 | **2.80** | 3.74 |
| iNat2018 | 2.70 | 2.40 | **1.69** | **1.48** | 0.90 | 0.28 | 0.72 | **1.21** | 1.51 |

