# OpenReview forum: "GeoBS: Information-Theoretic Quantification of Geographic Bias in AI Models"
_ICLR.cc/2026/Conference — Submitted to ICLR 2026_

### Official Review · Reviewer_vQQu · 2025-11-01

**Soundness:** 2
**Presentation:** 3
**Contribution:** 2
**Rating:** 4
**Confidence:** 4

**Summary:**

This paper proposes a mathematical frame-work for measuring the geographic bias (geo-bias) of predictive models. The frame-work asserts that there are three main aspects in the design space of geo-bias metric: the map, the reference pattern, and the difference measure, whose variations can construct different metrics. The paper also proposes new metrics of geo-bias that can capture bias related to multi-scalability, distance decay, and anisotropy, and provides experiments that utilize these metrics to measure the geo-bias of several foundation models.

**Strengths:**

The new metrics can be valuable for encouraging and enabling design of less geographically biased models. The insights about the existence of geo-bias in foundation models is also important for the users of these models in practice.

**Weaknesses:**

1. I find the mathematical frame-work uninformative. In particular, it is not clear where and how the frame-work can be useful. For example, the new metrics constructed by the paper do not stem from the theoretical framework, nor rely on it for any mathematical guarantees about their validity.

2. The figures in the paper lack clarity in my opinion, and particularly the captions are very uninformative.

3. The justifications for the use of KL divergence is not convincing. The first point is that it is less computationally expensive, but there are other linear-complexity options such as total variation. The second point about KL having a “physical” interpretation seems incorrect to me, providing reference for this claim is important (note that KL is not bounded and also is not a proper distance). Lastly, it is not clear why KL is chosen compared to reverse KL.

4. In Lines 374-377, the use of a binary metric for measuring continuous regression tasks seems unjustified, and the explanation of the binary metric lacks mathematical rigor.

5. The exclusion of some ROIs from evaluation, as pointed out in Lines 400-401, is not well-justified.

6. The choice of hyperparameters seems very ad-hoc, and it is unclear how changing these hyperparameters will affect the results and ranking of the models. This makes these metrics less useful in practice.

7. The proposed metrics, and the reported results, lack any measure of statistical significance (and a way to compute confidence intervals). Therefore, it is unclear how meaningful the differences between these metrics are.

8. The paper provides no validation for its proposed metrics for use in practical applications. How can a user trust conclusions drawn from the proposed metrics?

9. The paper draws some hypotheses from its analyses (Lines 440-444 and 466-472), but does not explore them any further. Providing some evidence to reject/affirm these hypotheses would be valuable for the development of future models.

Typos:

In Definition 4.3, “#” is undefined.

**Questions:**

1. Can the authors clarify the practical importance of the theoretical frame-work? Does it provide any guarantees, etc?
2. Can the authors provide any way to validate the metrics, for example, on synthetic data?
3. What are confidence intervals for the metrics?
4. What is the advantage of the proposed metric compared to performing independence test (chi-square on contingency tables) between model-correctness and geographical partitions?

---

> ### Author Response · Authors · 2025-11-26
>
> Dear Reviewer,
>
> Thank you very much for your comments. We hope our answers will address your concerns.
>
> Weakness 1. Our framework is based clearly on transforming classic spatial analysis theory concepts (e.g. unmarked and marked patterns, scale/distance/direction heterogeneity) into ML/DL compatible quantities, i.e., entropy/KL-divergence based loss functions. The goal is to 1) quantify geo-bias in a well-ordered, physically meaningful way, and 2) enable de-bias by minimizing the geo-bias scores as loss functions.
>
> Weakness 2. We are very sorry that we reduced the captions of Figures to the minimal to make the paper length meet the requirement. For example, in Figure (1), the dots show where an observation is evaluated by a model. Pure red dots mean 0 in prediction accuracy, and pure blue dots mean 1). The five examples of geo-bias scores may also look confusing to readers. We will improve our figures by 1) merging the examples into one region, one example, 2) stating clearly what the dots and colors mean, 3) explaining the concepts such as SSI and SRE in the caption.
>
> Weakness 3. Thank you for asking about the motivation for choosing KL-divergence. We'd like to clarify the misunderstanding here. 1) We say KL-divergence is "less computationally expensive" in comparison to Wasserstein distance. As we said in Weakness 1, we design our geo-bias scores to be **loss functions**. KL-divergence is commonly used in ML/DL as loss functions and proves to be very efficient. Total Variation (TV) of course is a valid candidate in term of linear complexity, but it is less computationally stable when used a loss function because of absolute values. 2) By physical, we mean that KL-divergence indicates the information loss if we compare the biased model performance to the "ideal", unbiased performance. The value of KL-divergence actually means some physical value compared to other heuristic measures, such as SPAD which is an ensemble of many experiments. 3) Reverse KL-divergence is mode-seeking, i.e. it is sensitive to a single mode, while forward KL-divergence is mass-covering. Considering that a ideal, unbiased performance map should look flat without obvious spikes, it is more reasonable to use forward KL-divergence than reverse KL-divergence.
>
> Weakness 4. Thank you for bringing this out. Our framework allows non-binary discretization, e.g. use quantiles/equal counts/equal intervals, to preserve more continuous information. A binary conversion is only a simple example to demonstrate how to use our geo-bias scores. The threshold we use in this paper to convert regression error into binary (good vs bad) is 1 sigma of the error population. It is a very common hyperparameter corresponding to the 68% threshold of the famous 68–95–99.7 rule of statistical significance testing. It roughly cuts the results of a regression model into 70% good and 30% bad, balanced for computation. If using other thresholds such as 2 sigma or 3 sigma, there will be only 5% or 0.3% bad cases, resulting in very small and unreasonable geo-bias measures.
>
> Weakness 5. As we have clearly stated in Lines 400-401, the reason we exclude ROIs that are "all 0s or all 1s" is that these scores will be *infinity*. There is no way to include them in computation. One alternative is to cap geo-bias scores with a very large value to avoid infinity. We can try this in the camera-ready version.
>
> Weakness 6. Please kindly refer to Line 406-421, where we stated our clear principles of choosing hyperparameters. It is based on the statistics of the dataset and there is no tuning of hyperparameters. For example, for a very sparse dataset, we need to set a larger radius r so that every neighborhood has a statistically sufficient amount of data (>30). We also reported a hyperparameter sensitivity test in Appendix B.3.
>
> Weakness 7. Thank you for pointing this out. We can run the models multiple times to get confidence intervals.
>
> Weakness 8 & 9. It is a very valuable suggestion that we test if our geo-bias scores can be used to discover the spatial tendency of large models as well as demonstrate the practical usefulness of our method. We wish to articulate that **knowing how to mathematically appropriately measure geo-bias" itself is already very important and useful, as no systemic geo-bias methods have been proposed, and model (especially foundation model) developers are mostly unaware of the potential risk of geo-bias (e.g. an LLM always gives wrong navigation suggestions in poor regions, causing safety issues). Let alone that our geo-bias scores can be used as loss function to do de-biasing.
>
> It is a very important future direction to explore the hypotheses we proposed in Lines 440-444 and 466-472. We are working on them as on-going projects.
>
> Q. 1-3: Please see our answers above.
>
> Q. 4: We provide a differentiable geo-bias measure which can be used as loss functions for de-biasing. Independence tests can not directly operate within the ML/DL framework.

---

### Official Review · Reviewer_y3HQ · 2025-11-01

**Soundness:** 3
**Presentation:** 3
**Contribution:** 3
**Rating:** 6
**Confidence:** 3

**Summary:**

This paper focus on the geo-bias evaluation in AI models, the authors propose GeoBS, an information-theoretic, model-agnostic, and spatially explicit framework for fair geo-bias evaluation. They further introduce three SRE scores considering multi-scalability, distance decay, and anisotropy. Experiments on 3 tasks, 8 datasets, and 8 models show that both GeoAI and several general-purpose models exhibit geo-bias, highlighting the need for spatial fairness in AI design.

**Strengths:**

1. I like this paper — combining spatial point pattern analysis with information theory for geo-bias evaluation is an innovative perspective that provides a theoretical lens. Although geographic bias might seem like a narrow topic within the broader fairness discussion, this kind of specialized, systematic, and in-depth evaluation has strong practical value.

2. This work successfully demonstrates how existing geo-bias metrics (Unmarked SSI and Marked SSI) can be interpreted within this framework, demonstrating good extensibility. The proposed metrics are model-agnostic and applicable to different types of AI models.

3. The spatially explicit nature makes evaluation results more interpretable. Experiments cover multiple tasks, datasets, and model types, enhancing the credibility of conclusions

**Weaknesses:**

1. The definition of "spatial homogeneity" is oversimplified; it's difficult to define what constitutes an "unbiased" reference pattern in practice.
2. Lacks sensitivity analysis for different reference pattern choices; choice of KL divergence lacks sufficient justification.
3. Only considers first-order statistics, potentially missing important spatial interaction information.
4. Hyperparameter choices (ROI radius, grid size, ...) significantly impact results, but lack systematic selection guidance.
5. Lacks statistical significance testing, making it difficult to judge whether observed differences are statistically meaningful.

**Questions:**

1. How should appropriate "unbiased" reference patterns be selected for different application scenarios? Can you provide a  guiding discussions?
2. Can you provide some discussions showing how these bias scores could guide practitioners in improving their models?
3.  SRE scores themselves show high hyperparameter  sensitivity. How will you address this issue?

---

> ### Author Response · Authors · 2025-11-26
>
> Dear Reviewer,
>
> Thank you very much for your insightful comments and encouragements. We hope our brief discussions can help address your questions.
>
> Weakness 1. The definition of "spatial homogeneity" is oversimplified; it's difficult to define what constitutes an "unbiased" reference pattern in practice.
>
> Answer: Thank you for pointing this out. In practice, we define "unbiased" reference pattern as "a randomly generated noise". As to what distribution we generate the noise from depends on domain knowledge and personal discretion. For example, if we are measuring a POI visiting model, it is appropriate to use a spatial point Poisson process to generate the homogeneous pattern.
>
> Weakness 2. Lacks sensitivity analysis for different reference pattern choices; choice of KL divergence lacks sufficient justification.
> Only considers first-order statistics, potentially missing important spatial interaction information.
>
> Answer: We reported a sensitivity test in Appendix B.3. By comparing Table 6 and Table 7, we can draw a preliminary conclusion that SSI scores are much more stable than SRE scores.
>
> We are motivated to choose KL-divergence because: 1) It is an efficient measurement of distribution difference, less computationally expensive (e.g. compared to Wasserstein distance) and more computationally stable (e.g. compared to total variation). 2) We design our geo-bias scores to be loss functions so that we can do de-biasing in the future. KL-divergence is commonly used in ML/DL as loss functions and proves to be very efficient. 3) KL-divergence can be clearly interpreted. In information theory, KL-divergence indicates the information loss if we compare the biased model performance to the "ideal", unbiased performance. The value of KL-divergence actually means some physical value compared to other heuristic measures, such as SPAD which is an ensemble of many experiments.
>
> It is a very good idea to include second-order statistics in geo-bias measurement. We will explore this possibility in our future work.
>
> Weakness 3. Hyperparameter choices (ROI radius, grid size, ...) significantly impact results, but lack systematic selection guidance.
>
> Please kindly refer to Line 406-421, where we stated our principles of choosing hyperparameters. In short, the choice of radius/scale/distance/slice both depends on what spatial aspects you want to study (e.g. a city-scale problem should use a scale smaller than 1mile) **and** statistical stability (e.g., we usually require radius to be chosen large enough so that each neighborhood has at least 30 samples). We also reported a hyperparameter sensitivity test in Appendix B.3.
>
> Weakness 4. Lacks statistical significance testing, making it difficult to judge whether observed differences are statistically meaningful.
>
> Thank you for pointing this out. We can run the models multiple times to get confidence intervals. We did not report this mostly due to resource limits -- we only have access to pretrained weights of many of the models and we are unable to generate multiple-run statistics unless we can re-train them. We can try to add some noise to the inputs so that we get different results for each experiment and report the statistical significance.
>
> Q1. How should appropriate "unbiased" reference patterns be selected for different application scenarios? Can you provide a guiding discussions?
>
> We propose to design unbiased reference patterns based on domain knowledge. For many subfields in spatial modeling (e.g. POI visiting, hydrology, natural hazards), there are accumulated data whose average patterns can be considered natural and unbiased.
>
> Q2. Can you provide some discussions showing how these bias scores could guide practitioners in improving their models?
>
> We can see two immediate benefits of adopting geo-bias scores:
>
> 1) It can be used to evaluate models, especially large foundation models, more comprehensively. Most model (especially foundation model) developers are unaware of the potential risk of geo-bias, for example, **an LLM always gives wrong navigation suggestions in poor regions, causing safety issues**, or **generating positive reviews on Yelp for western-style restaurants and negative reviews for African foods**. LLM researchers used to fix the bad cases, but they lack a systemic measure of how geo-biased their models are.
>
> 2) We design geo-bias scores to be loss functions. They can be used for de-biasing -- i.e. add geo-bias score to the model's training loss function to reduce geo-bias from its root. The de-biasing experiments are part of our working paper so we are unable to include them in this paper's scope.
>
> Q3. SRE scores themselves show high hyperparameter sensitivity. How will you address this issue?
>
> That is a very good question. We are considering an alternative: we calculate a set of SRE scores with different hyperparameters and report the most abrupt change. That may help reduce the sensitivity.

---

### Official Review · Reviewer_EaiB · 2025-11-03

**Soundness:** 2
**Presentation:** 2
**Contribution:** 2
**Rating:** 2
**Confidence:** 3

**Summary:**

The paper introduces an information-theoretic and model-agnostic framework for quantifying geographical bias. This framework can integrate existing metrics and proposes three new metrics that account for multi-scalability, distance decay, and anisotropy. The authors perform experiments across multiple models to reveal that there exist geobias in both task-specific geoAI models and foundation models.

**Strengths:**

The proposed framework is model-agnostic, addressing a key limitation of prior work, which tends to be restricted to specific tasks. They include ways to quantify bias by including intricate details about space, distance, and scale. This ensures the quantifying of bias in a granular way.

**Weaknesses:**

In my opinion, the abstract of the paper is misleading because of the part: “ but will also establish a foundation for integrating spatial fairness into the design, deployment, and evaluation of AI systems”. The framework is purely diagnostic, and no mitigation scheme is mentioned.

Figure 1 is a very important figure that helps motivate the importance of quantifying bias in different ways. But it's hard to understand what the different colored points represent since they are not all explicitly mentioned. This makes the overall message of the figure unclear. Additionally, using the same spatial area to illustrate each metric would make the differences between them easier to compare and understand. The acronym SSI is not defined and explained at this stage (it is done way later in the paper). It would have been to see if the authors described what they meant by geo bias clearly with some easy-to-understand intuitive examples.

The writing of the paper is occasionally difficult to follow. For instance, “Spatial point patterns always involve multiple locations (a single point will not form 'patterns'), so we define the unit to evaluate geo-bias as.”. This is followed by a series of undefined notations (e.g., what i,m,s, and t index). The first paragraph of Section 4 is missing logical flow. It starts off by mentioning the objectives, then explains how it serves the purpose. Where the purpose itself is not mentioned. Additionally, they state “three key factors we need to consider when designing new geo-bias metrics”, what are those key factors? It's very important to clarify the differences between unmarked SSI and marked SSI. Figure 2 can be a little misleading since two different locations are used to visualize the geo-bias scenarios. The main idea is that they would both have the same number of points, except one would just have the points against the unobserved background, and the other would have the same points with different colors representing high and low performance. The placement of Table 1 is rather odd, I would want to have it near the intro, not near the end, when I'm almost done reading. And these acronyms are used throughout the texts, not just in the following section.

For definition 4.3, having #P_k/#N outside the sum function makes no sense as to what the definition intends. They are supposed to be using the weight for each patch, not just one.

Converting regression values to binary might lose valuable information, and the choice of threshold should be justified.

Since SPAD is used as a baseline, details about what exactly it quantifies should be mentioned.

The claim that “The geo-bias scores are significantly lower than the task-specific counterparts” is not supported by the presented tables. While Table 4 shows lower bias values relative to Table 3, the comparison to Table 2 contradicts this assertion: many entries in Table 4 are comparable to, or higher than, those in Table 2. The authors should either (a) clarify exactly which comparisons support the “significantly lower” claim, (b) report formal significance tests (with effect sizes and confidence intervals) for the claimed differences, or (c) revise the statement to accurately reflect the mixed results.

The parameter sensitivity analysis is conducted only for a limited subset of models and datasets, without any justification for their selection. Without explaining why these particular models and datasets were chosen, the analysis cannot be considered a comprehensive or reliable assessment of parameter tuning.

The paper does motivate the importance of a model-agnostic approach and the importance of spatial info like (distance, direction, and scale). The discussion remains at a general level, reporting the presence of bias without interpreting its variation by scale, distance decay, or directional (anisotropic) patterns. Providing such an interpretation would strengthen the link between the theoretical motivation and the empirical findings.

In Figure 3, are the bars on the left for a single patch? The caption itself is not informative enough, and there's no mention of this in the text.
Details about the hyperparameter tuning of the models are also missing.

**Questions:**

In addition to the weaknesses discussed earlier, I would appreciate the authors' responses on the following:

* In Figure 3, are the bars on the left for a single patch?

* Why was the parameter sensitivity analysis conducted only for a limited subset of models and datasets?

*Why was GPT-4o chosen as one of the fms?

---

> ### Author Response · Authors · 2025-11-26
>
> Dear Reviewer,
>
> Thank you very much for your detailed and insightful comments. We'd like to briefly explain the confused points and hope it will address your concerns.
>
> (1) Lack of mitigation. We apologize for the confusion here. In this paper, we proposed 5 distinct geo-bias metrics that are **deliberately designed as entropy or KL-divergence**, which allows them to be used directly as auxiliary loss functions. By "mitigation" we mean that users can add geo-bias scores to their own loss functions and reduce their model geo-bias on the fly. The de-biasing experiments are part of our working paper so we are unable to include them in this paper's scope.
>
> (2) Ambiguity of Figure (1). We are very sorry that we reduced the caption of Figure (1) to the minimal to make the paper length meet the requirement. The dots show where an observation is evaluated by a model, and the more red, the worse the performance (more specifically, pure red means 0 in prediction accuracy, and pure blue means 1). All five subfigures are further explained in Figure 2 and Figure 3. We will add more details to the caption of Figure (1), including the concept of SSI, in the camera-ready version.
>
> (3) Writing styles. Again, we apologize for the readability issues, especially regarding the confusion on SSI. Most of the "logic gaps" you mentioned (e.g. three key factors, the unexplained purpose of designing objectives) are domain knowledge of spatial analysis (e.g. distance, direction and scale are the key factors of analyzing a spatial event). We will explicitly add these missing background information in the camera-ready version. SSI is a fairly complicated spatial analysis concept proposed in the literature. We will bring up this concept earlier in the paper and dedicate a section in the Appendix to give more background information in the camera-ready version.
>
> We are a bit uncertain what you exactly mean by "Spatial point patterns ... evaluate geo-bias as. This is followed by a series of undefined notations (e.g., **what i,m,s, and t index**)." We went back to the manuscript, where followed by the statement in Line 148-149 are Line 150-157, three clear Definitions 3.5, 3.6, 3.7. **There are no i, s or t indices**. Are you referring to other sections of the paper?
>
> (4) Error in Definition 4.3. Thank you very much! It is a typo. The #P_k/#N should be inside the sum. We will correct it.
>
> (5) Converting regression values to binary. Thank you for bringing this out. Our framework allows non-binary discretization, e.g. use quantiles/equal counts/equal intervals, to preserve more continuous information. The threshold we use to convert regression error into binary (good vs bad) is 1 sigma of the error population. It is a very common hyperparameter corresponding to the 68% threshold of the famous 68–95–99.7 rule of statistical significance testing.
>
> (6) SPAD. Briefly speaking, SPAD is cutting the space into many different patches and calculate their model performance variation across patches. A perfectly uniform model should yield 0 variation. We will add a section in the Appendix to give more background information in the camera-ready version.
>
> (7) "The geo-bias scores are significantly lower than the task-specific counterparts" claim. We are sorry for the confusion -- we mean the geo-bias of SatMAE compared to other task-specific counterparts. We corrected the statement in the manuscript.
>
> (8) Hyperparameter sensitivity. The principles we use to choose hyperparameters are stated in Line 406-421. It is based on the statistics of the dataset and there is no tuning of hyperparameters. For example, for a very sparse dataset, we need to set a larger radius *r* so that every neighborhood has a statistically sufficient amount of data (>30).
>
> There is misunderstanding on the models we choose to do the sensitivity test. Please see Table 2 and Table 3 -- rbf, Space2Vec, NeRF and Sphere2Vec are exactly the 4 models we used in the sensitivity test Appendix Table 6. The Space2Vec-Theory and Sphere2Vec-SphereC are typos. We choose iNat2018 as the dataset because it is the **largest** and has the best spatial coverage. We can report the results on all datasets in the camera-ready version.
>
> (9) Importance of spatial factors in bias. As we explained, there are widely recognized principles in spatial analysis and GIS -- we will add more literature to the camera-ready version for better readability.
>
> (10) Figure (3). Sorry for the confusion. Yes, Figure (3) illustrates an example of one patch vs the entire ROI.
>
> (11) Our lab only has GPT-4o quota at the time of experiments. We can add more LLM results in the camera-ready version.

---

### Author Response · Authors · 2025-12-04
**Summary of Rebuttal**

Dear Area Chair, Senior Area Chairs, Program Chairs

Due to the reversion of the rebuttal process, we provide this concise summary of our contribution and our response to the reviewers' concerns.

As Reviewer y3HQ (rating 6) states, the strengths and values of our paper are:

- Combining spatial point pattern analysis with information theory for geo-bias evaluation is an innovative perspective that provides a theoretical lens.

- We demonstrate how our framework interprets existing geo-bias metrics (Unmarked SSI and Marked SSI, NeurIPS 2024) and extends to new metrics (SREs), applicable to a wide range of AI models.

- The spatially explicit nature makes evaluation results more interpretable.

Our work is a systemic summary of previous geo-bias evaluation work and guides the design of new geo-bias metrics. The information-theoretic interpretation guarantees that it is not an ad hoc, model-and-task-specific solution. The ties with classic GIS theory makes it interpretable to domain scientists. It bridges general AI evaluations with the specific needs of geo-tasks. Many large models, while being accurate overall, performs in a spatially biased way without arousing users' awareness because there are no systemic geo-bias metrics. Our work has very realistic impacts.

In general, there are three sets of questions:

1. **Clarification on writing**. Most questions of Reviewer EaiB (rating 2) are about clarification on certain statements in the manuscript. The reviewer raised concerns about figure caption being too brief, domain concepts such as SSI and SPAD lacking detailed introduction, typos in definitions and etc. Reviewer vQQu (rating 4) raised similar questions about figure captions. We sincerely appreciate these comments and edited our manuscript accordingly to address all the concerns.

2. **Lack of domain knowledge background**. Reviewer EaiB (rating 2) and Reviewer vQQu (rating 4) seemed to be unfamiliar with some of the GIS/spatial analysis domain concepts and misunderstood our motivations/justifications. For example, the importance and necessity to consider scale, distance and direction factors in geo-bias evaluation is based on classic spatial analysis principles. We added more background discussions on these concepts in the Appendix.

3. **Technical details**. For example, both Reviewer EaiB (rating 2) and Reviewer vQQu (rating 4) asked for more details on hyperparameter sensitivity tests and regression discretization. We pointed the reviewers to the descriptions in the manuscript that answer their questions. Reviewer vQQu (rating 4) asked why we used KL-divergence, which we articulated in the manuscript -- for the potential of loss functions.

We argue that we have addressed the weaknesses raised by the reviewers, most of them being concerns about writing. We sincerely appreciate these comments which helped us improve the flow of our paper. **However, we believe that these weaknesses do not harm the core contributions and novelty of our research.** We respectfully ask that the AC consider the intellectual merits and broader impacts of our work. We would like to thank all the reviewers again for taking time and raising important questions helping us improve the quality of our work.

Sincerely,

The Authors.

---

### Meta-Review · Area_Chair_3Agp · 2025-12-31

**Summary:**

The authors study spatial bias detection. This is an important, little-studied area bordering trustworthy AI and spatial data science. Unfortunately, none of the reviewers has been enthusiastic about the paper. Key issues include:

- The writing is unclear in places and lacks mathematical rigour. Specifically, the key notion relies on "unbiased" reference patterns, whose design is unclear. In the rebuttal, authors suggest "We propose to design unbiased reference patterns based on domain knowledge. For many subfields in spatial modeling (e.g. POI visiting, hydrology, natural hazards), there are accumulated data whose average patterns can be considered natural and unbiased," which seems somewhat underwhelming.

- Lack of domain knowledge: The related-work section and experimental evaluation are limited. The comparison focuses on TorchSpatial:
https://proceedings.neurips.cc/paper_files/paper/2024/file/9449c2d5b0cc8c9a445752f3ff195a1c-Paper-Datasets_and_Benchmarks_Track.pdf
while disregarding much of the alternatives, incl.:
https://arxiv.org/abs/2302.12333
https://openproceedings.org/2025/conf/edbt/paper-263.pdf
https://dl.acm.org/doi/abs/10.1145/3748636.3762760

 - There are no tests of statistical significance of the results reported.

**Reviewer Concerns:**

In the rebuttal to the more critical reviewers (e.g. vQQu), the authors have just acknowledged the weaknesses.

**Reviewer Scores:**

I doubt that any of the reviewers would improve their scores.

---

### Decision · Program_Chairs · 2026-01-26

Reject